**Subject Category:**
Biology (whole organism)

health and disease and epidemiology

cattle, contact chains, epidemiology, network analysis, superspreader

**Author for correspondence:**
Robbie A. McDonald
e-mail: r.mcdonald@exeter.ac.uk

# Contact chains of cattle farms in Great Britain

Helen R. Fielding[1], Trevelyan J. McKinley[2], Matthew J. Silk[1], Richard J. Delahay[3] and Robbie A. McDonald[1]

[1]Environment and Sustainability Institute, and [2]College of Engineering, Mathematics and Physical Sciences, University of Exeter, Penryn Campus, Penryn TR10 9FE, UK
[3]Animal and Plant Health Agency, Woodchester Park, Nympsfield, Stonehouse GL10 3UJ, UK

HRF, 0000-0001-7041-1380; TJM, 0000-0002-9485-3236;
MJS, 0000-0002-8318-5383; RJD, 0000-0001-5863-0820;
RAM, 0000-0002-6922-3195

Network analyses can assist in predicting the course of epidemics. Time-directed paths or 'contact chains' provide a measure of host-connectedness across specified timeframes, and so represent potential pathways for spread of infections with different epidemiological characteristics. We analysed networks and contact chains of cattle farms in Great Britain using Cattle Tracing System data from 2001 to 2015. We focused on the potential for between-farm transmission of bovine tuberculosis, a chronic infection with potential for hidden spread through the network. Networks were characterized by scale-free type properties, where individual farms were found to be influential 'hubs' in the network. We found a markedly bimodal distribution of farms with either small or very large ingoing and outgoing contact chains (ICCs and OCCs). As a result of their cattle purchases within 12-month periods, 47% of British farms were connected by ICCs to more than 1000 other farms and 16% were connected to more than 10 000 other farms. As a result of their cattle sales within 12-month periods, 66% of farms had OCCs that reached more than 1000 other farms and 15% reached more than 10 000 other farms. Over 19 000 farms had both ICCs and OCCs reaching more than 10 000 farms for two or more years. While farms with more contacts in their ICCs *or* OCCs might play an important role in disease spread, farms with extensive ICCs *and* OCCs might be particularly important by being at higher risk of both acquiring and disseminating infections.

## 1. Introduction

Pathogen transmission among hosts may occur by a variety of routes, from different types of direct contact, to indirect contact via vectors, fomites and the environment [1]. For livestock, animal movements between farms can be considered to form a

directional link from the source to the destination farm, which may therefore indicate potential pathways for direct and indirect transmission of pathogens [2–5]. Conceiving of farms as nodes and animal movements as edges in network analyses has been well developed in theory [6], and has been applied to networks of holdings in multiple livestock species [7–12]. Centrality measures can indicate the importance of a given farm within a trading network [13], and preferential protection, treatment or isolation of more central, or more influential, farms might enhance disease control measures [9,14,15]. Network measures such as a farm's *degree* and *strength* (the number of other farms with which they trade and the number of animals traded, respectively) have been associated with infection risks [16].

In a structured population, the transmission of infection depends on the frequency and nature of interactions among individuals and groups. Cross *et al*. [17] noted that for a fixed frequency of movements, an acute disease with a short infectious period encountering a sparse network will be unable to spread extensively before extinction. However, a chronic disease in the same network might be able to persist and disseminate more widely, if it has a long infectious period, relative to the frequency of between-group interactions [17]. Therefore, investigation of such networks requires consideration of the temporal aspects of infectiousness of the pathogen, relative to the frequency of movements [18]. Static network analyses can be particularly useful in evaluating disease transmission where this period of risk is quantified on the same temporal scale as the network [19–21]. For example, in studying transmission of foot and mouth disease virus (FMDV), which has an incubation period of a few days [22], networks encompassing one week of movements are appropriate. But, for a chronic infection such as bovine tuberculosis (bTB), infection can be asymptomatic or latent for months to years [23,24], and a longer-term perspective is required. In reality, the edge of a static network permanently depicts what is truly a transient event, persisting as long as infection does, either in the incoming animal, a contaminated environment, or via secondary infections in other livestock or wildlife. It has been shown, in networks constructed from movements of adult dairy cows, that analysis of groups of statically connected nodes consistently overestimated the epidemic size of highly transmissible diseases, whereas measures that took into account the temporal order of movements provided a lower and more realistic estimate [25,26]. The concept of contact or infection chains has been reasonably well developed in the field of veterinary epidemiology [7,27,28] and the wider literature contains many examples of essentially the same approach, such as *time-directed paths* [29], *source counts* [30], *accessible worlds* [31], *output domains* [25] and *reachability* [8,32,33]. All of these terms describe a temporally sequential network to identify the nodes that are accessible through edges to/from each index node within a selected time period. In this study, we have used the term *contact chain* for consistency with the current literature in our field. Ingoing contact chains (ICCs) identify the number of farms that could *potentially* transmit infection to the index farm over a defined period arising from the purchase and importation of animals. Outgoing contact chains (OCCs) quantify the number of 'downstream' farms that could *potentially* acquire infection from the index farm through its onward sale and export of animals. This structure of contacts may therefore help to predict the risk of the index farm acquiring and then passing on infection and to characterize patterns of risk across a national herd. Of course, not all movements result in the transmission of infection; at least one animal moved per edge must be infected and have the prospect of becoming infectious, to have a chance of infecting animals on other farms down the chain. Crucially, in our study we do not explicitly model transmission of infection and we use the term 'contact' chain, rather than 'infection' chain, representing only the *potential* for infection spread. For infections that can effectively be clinically hidden, such as bTB, contact chains can provide a scale, extent and map of potential transmission routes, which may improve our understanding of epidemiology beyond that available through studying direct contacts. In a 5-year study of the French cattle movement network, where bTB is rare (relative to the UK), farms in the highest quartile of ICCs traded indirectly with up to 84% of farms in the network [34]. It was shown that these farms in the highest quartile of ICCs were more likely to experience a bTB outbreak [34], suggesting a link between the connectedness of farms through the purchase of animals and their risk of acquiring this chronic infection. Similarly, the magnitude of ICCs has been associated with the risk of acquiring an acute infection, bovine coronavirus, on Swedish cattle farms [35].

The cattle industry in Great Britain relies heavily on trade in animals among beef and dairy producers. Trading occurs privately, through a dealer or via livestock markets [36] and each movement of a bovine animal is recorded by a national Cattle Tracing System. These records have been used to study both network structure and cattle demographics [37]; Green *et al*. [38] analysed the initial spread of the FMDV outbreak in 2001, before movement restrictions were implemented, revealing that livestock movements could result in widespread dissemination of the virus and that the timing of virus introduction affected epidemic spread through seasonal fluctuations in movements

among farms. The susceptibility of a network to infection also depends on its overall connectivity (i.e. how many sections or components into which it is divided). Heterogeneity in British cattle movements is predicted to influence disease spread [39] and so we have looked for known characteristics of farms that align with their trading behaviours. Production type and herd size have been found to be important in predicting movements among pig [40,41] and cattle farms [7,10] and have been associated with the persistence of bTB [42]. We predicted variation in network measures and contact chains, based on herd size, production type and location, and thereby we expect that they might have varying influences on potential transmission of infection. We use both static and temporally relevant network analyses in the context of a chronic livestock disease in Great Britain to provide insight into the dynamics of cattle trading behaviour, investigate potential unobserved transmission routes and to characterize the important actors and practices within this network. We performed our network and contact chain analyses annually over an extended period to determine if changes within the British cattle industry have been reflected in the network structure or individual farm behaviour over time, and consequently if they might have the potential to affect disease transmission within the cattle population.

# 2. Methods

## 2.1. Population-level analysis

The Cattle Tracing System (CTS) records all movements, births and deaths of British bovines. For our study, the Animal and Plant Health Agency provided a cleaned, processed version of CTS data (see [43]) on the recorded movements of cattle between locations in Britain from 1 January 2001, when recording became mandatory [43], to 31 December 2015. Data consisted of 158 million individual animal movements between premises. We removed births (41 million) and deaths (42 million) from the dataset and aggregated individual animal movements into batch movements of animals moved between the same farms on the same day. We included only animal holdings (farms) in this study, omitting 34 million movements to slaughterhouses as they represent sinks in the network where no epidemiologically significant transmission could occur. Twenty-six million movements (35%) took place via markets or show grounds; we classed these as transitory and linked them as single edges from source to ultimate destination, removing the transitory node. Although we acknowledge the well-documented risks of livestock mixing at showing events and markets [44], and a market's role in concentrating and dispersing animals [36], we considered premises that kept animals for longer than one day to be more relevant for transmission and persistence of slow-spreading infections, such as bTB. By directly linking the source and ultimate destination, the flow of animals through these premises remains in the analysis, while allowing us to focus on the farm premises, upon which opportunities for transmission of infection were most prolonged. We took 12-month periods from 1 January to 31 December for each year between 2001 and 2015 and grouped batch movements into single links between farms. These processes together left 9.5 million edges in the study network. Annual herd size was calculated as the mean of the daily number of animals on the farm over the same 12-month period. We used CTS data to define herd type for each year, defining it by the predominant classification (beef, dairy or dual purpose) based on breed and then predominant sex within this classification. *Suckler* farms were defined by a majority of female beef animals, aiming to capture those herds where calves are reared by their dams before weaning (cow-calf systems). *Dairy* farms were defined by a female dairy majority, identifying herds producing milk commercially. *Fattening* units were defined by a male animal majority, identifying herds that did not breed cattle, but reared them for beef production. Any farms where the breed type or subsequent sex was not more than 50% were defined as *mixed herds*.

## 2.2. Network analysis

We constructed networks in which nodes were defined as unique animal holdings registered as keeping cattle, and directed edges were defined as a movement of one or more cattle between holdings. Directed edges were weighted by the number of animals moved to/from the same holdings during the network year, as we considered the number of animals to be proportional to the risk of a disease incursion, especially for a disease with low prevalence within herds [15]. Only active holdings (those with a recorded movement, birth or death in the year of study) were included in each annual network. The

network timeframe corresponded with previously defined 12-month periods between 2001 and 2015. Using a full year avoided bias from seasonal variation of movements [45], yet was sufficient for transmission of a chronic infection [20]. In- and out-degree, in- and out-strength, betweenness, edge density, degree assortativity, reciprocity, clustering coefficient, average path length, and the giant weakly and strongly connected components (GWCC and GSCC) were all calculated using the R [46] package igraph [47]. Definitions of all network measures and accompanying functions are provided in electronic supplementary material, table S1. We compared measures from the observed networks in each year to values calculated from directed random networks of the same size and density, generated using the Erdös–Renyi model [48] (see electronic supplementary material, Methods).

## 2.3. Contact chains

We calculated ICCs and OCCs for 1-year periods (starting and ending in January) from 2001 to 2015 using the R package 'EpiContactTrace' [49]. Overall, the effect of seasonal variations in movement patterns [45] on our contact chains was likely to be minimal as we used a whole year of movements. However, should a farm have purchased a large group of animals in January every year and calculation of their chains began at the beginning of January, the chains would never have a chance to 'build', as the incoming movements to the farms from which they had purchased would not be included. Therefore this farm's 'true' chain would only be apparent in a chain that started in a different starting month, e.g. December. Any such effect could result in underestimation of the magnitude of the chain for some farms. Therefore, we calculated ICCs and OCCs starting at consecutive monthly intervals from January 2012 to December 2013, a total of 24 1-year periods (see electronic supplementary material, figure S1 for schematic). We compared the results from the different starting months and then combined these 24 1-year periods to create a more robust summary of movements spanning 36 months (from the start of the earliest chain to the end of the latest), rather than a 12-month snapshot. We compared the mean, median and maximum number of farms from the combined 24 monthly spaced chains and combined annually spaced chains over the same time period (2012–2014) using Spearman's rank analysis. Summary values from both methods were very similar (see electronic supplementary material), and so our subsequent analysis used the mean of the 24 monthly spaced chains. For comparison of chains 2001–2015, we used single 12-month chains to reduce computational load. We set thresholds on a logarithmic scale for the number of farms in chains to aid their description; 0–10 = very small, 11–100 = small, 101–1000 = intermediate, 1001–10 000 = large, more than 10 000 = very large. The maximum number of farms in a contact chain in any 1 year represents the greatest extent of the potential impact any one farm may have on the network in that year. Correlations between network measures were randomized to account for non-independence of network data. Temporal stability of measures was assessed through ranking of nodes and calculating the standard deviation of mean rank over time [50] (see electronic supplementary material).

To characterize those farms at high risk of acquiring infection (defined as farms with very large ICCs) or of spreading infection (defined as farms with very large OCCs), or potential 'superspreaders' (defined as farms with very large ICCs and very large OCCs), we used a logistic regression with a binomial error structure. We performed the analysis using a threshold of 100, 1000 and 10 000 farms (see electronic supplementary material for ROC values and predicted probabilities). The highest ROC values were achieved using a threshold of 10 000 farms and so this was used for the final models. Herd type, size and region have previously been associated with contact chains of cattle farms in Sweden and Uruguay [7,10] and were therefore used in our logistic regression. We grouped Great Britain into 10 regions for this analysis (electronic supplementary material, figure S10). We tested the full model using backward stepwise selection based on Akaike's information criterion (AIC) [51] but found in every case that the full models had the lowest AIC. We calculated odds ratios and confidence intervals and performed ROC curve analysis to estimate the model goodness of fit [52].

# 3. Results

## 3.1. National herd characteristics

The median number of cattle traded from a single farm to another over the 12-month period was 2 (interquartile range 1–4), and this remained similar across all years, apart from in 2001 when larger numbers of cattle were moved between farms (median = 3, interquartile range 1–8, max = 3990). Most cattle holdings (mean = 40 736, 56.8% of all farms) were characterized as suckler herds

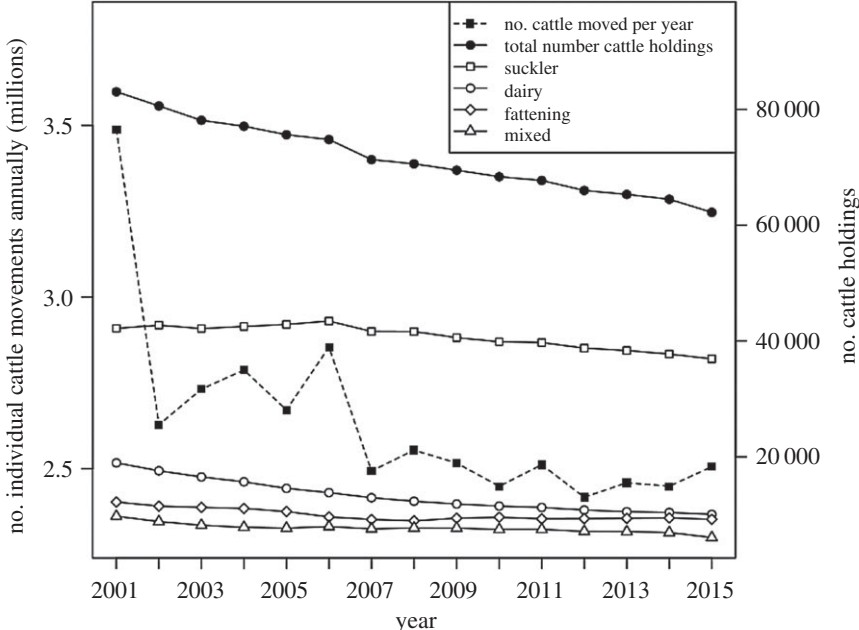

**Figure 1.** Numbers of cattle moved between animal holdings (cattle farms) and the number of holdings, characterized by herd type, in Great Britain from 2001 to 2015. Data from the Cattle Tracing System. The numbers of cattle moved are shown with a dashed line and the numbers of holdings are shown with solid lines.

(figure 1). The total number of cattle holdings (number of nodes in the network for which we had sufficient information to classify the herd type) decreased by 20 840 during the study period, and this reduction was most evident among dairy herds (figure 1 and table 1). Herd sizes increased over the study period; dairy herd size increased from a median of 166 in 2001 to 243 in 2015, while the numbers of animals in other herd types remained more stable (electronic supplementary material, figure S2). The predominant between-herd flows of cattle were among suckler herds (12%), from suckler to fattening herds (9%), and from dairy to fattening herds (7%) (figure 2a). Suckler and dairy farms were the source for 23% and 18% of movements, respectively, between 2012 and 2014, despite the number of dairy farms being approximately a quarter that of suckler farms. Fattening farms received the most (21%) movements from other farms and dairy farms received the fewest (4%), of which most were from other dairy herds. Thirty-five per cent of movements were traded through markets, often from breeding herds (suckler and dairy) back to suckler herds or to fattening herds (figure 2b). The number of active farms (those with a birth, death or movement) stayed stable after 2001 (table 1). 'Isolated', farms with a birth or death but *not* participating in the network make up 6.9% of farms (table 1). On average, 34.4% of dairy holdings purchased no cattle in any one year (95% confidence interval = 31.8–37.1%), followed by suckler farms at 27.0% (23.9–30.1%), mixed holdings at 22.5% (19.1–25.7%) and fattening units had the lowest percentage of closed farms at 9.64% (7.5–11.8%). Farms that did not report any inward movements in a 5-year period (2011–2015), and so could be considered 'closed' within this time, made up 9.6% of herds in the network. This is a similar value to dairy, suckler and mixed herds at 10.1%, 10.5% and 10.6%, respectively. We found 5.4% of herds categorized as fattening enterprises were 'closed' for that period, suggesting that these were not typical 'fattening' units and that more than one form of enterprise was present.

## 3.2. Network analysis

The number of edges (created when at least one animal is traded between two farms) in the cattle movement network was lowest in 2001, after which it stabilized to between 60 000 and 70 000 between-herd links per annum (table 1). Every year, all metrics of the observed network values lay outside the distribution of values from the random networks (electronic supplementary material, table S2), showing that all of the observed networks differed from random in a range of key measures. The density of connections between farms in the network (edge density) increased by over 50% during the study period (table 1). Degree assortativity was more negative in all years in the observed network,

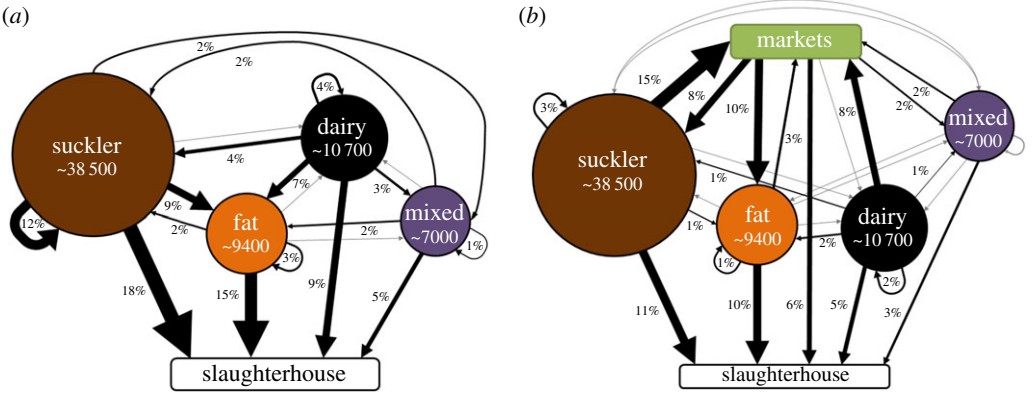

**Figure 2.** Relative importance of pathways for cattle movements in Great Britain from 1 January 2012 to 31 December 2014. Movements are shown as a percentage of 13 910 851 movements over this period. Black arrows represent movements weighted as a percentage of total movements recorded. Flows of less than 1% are marked by grey arrows. The number of farms in each herd type is the mean number of unique, registered cattle holdings over the years 2012–2014, also represented by the area of the circle for each type. Our analyses are based on the characterization of movements shown in (*a*) which represents cattle movements among cattle holdings (farms) and slaughterhouses, and where movements made through markets are included as direct farm-to-farm/slaughterhouse movements. For comparison, in (*b*), markets are included explicitly as locations, to indicate the frequency of cattle movements via markets.

**Table 1.** Global network analysis metrics for observed directed, weighted networks of cattle movements between animal holdings (farms) in Britain from 2001 to 2015. Data are from the Cattle Tracing System.

| year | nodes (farms in network) | edges (movements of at least one animal between two farms) | active farms (% of total farms) | edge density | reciprocity (0–1) | clustering coefficient (0–1) | average path length (no. of steps) |
|---|---|---|---|---|---|---|---|
| 2001 | 85 410 | 444 514 | 95.2 | 0.000061 | 0.0319 | 0.0035 | 6.54 |
| 2002 | 82 916 | 585 262 | 93.4 | 0.000085 | 0.0351 | 0.0111 | 7.14 |
| 2003 | 80 428 | 694 775 | 92.7 | 0.000107 | 0.0368 | 0.0138 | 6.45 |
| 2004 | 79 404 | 707 915 | 92.7 | 0.000112 | 0.0397 | 0.0150 | 6.42 |
| 2005 | 77 800 | 674 776 | 93.2 | 0.000111 | 0.0422 | 0.0144 | 6.54 |
| 2006 | 76 970 | 733 241 | 93.7 | 0.000124 | 0.0382 | 0.0138 | 6.33 |
| 2007 | 73 380 | 622 011 | 92.2 | 0.000116 | 0.0379 | 0.0139 | 6.49 |
| 2008 | 72 624 | 650 563 | 93.0 | 0.000123 | 0.0353 | 0.0150 | 6.57 |
| 2009 | 71 485 | 668 151 | 93.3 | 0.000131 | 0.0339 | 0.0146 | 6.41 |
| 2010 | 70 328 | 639 541 | 93.1 | 0.000129 | 0.0342 | 0.0143 | 6.60 |
| 2011 | 69 649 | 647 205 | 93.3 | 0.000133 | 0.0329 | 0.0144 | 6.67 |
| 2012 | 67 820 | 625 378 | 92.9 | 0.000136 | 0.0315 | 0.0138 | 6.62 |
| 2013 | 67 171 | 616 215 | 93.0 | 0.000137 | 0.0305 | 0.0134 | 6.97 |
| 2014 | 66 292 | 616 627 | 93.0 | 0.000140 | 0.0303 | 0.0136 | 6.93 |
| 2015 | 64 624 | 619 632 | 92.0 | 0.000148 | 0.0306 | 0.0132 | 6.89 |
| mean | 73 753 | 636 387 | 93.1 | 0.000120 | 0.0347 | 0.0132 | 6.64 |

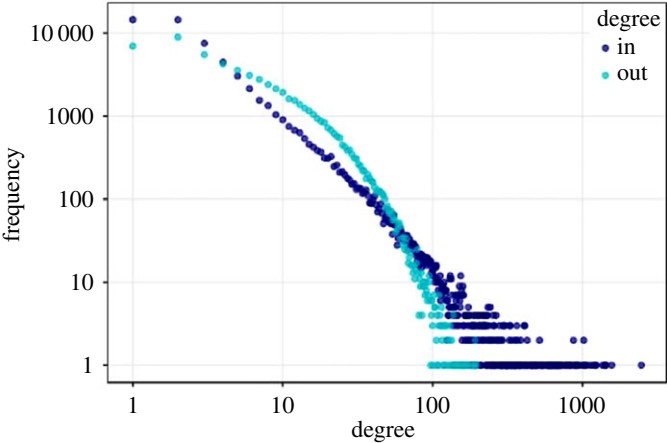

**Figure 3.** Distribution of in- and out-degree for all farms in the annual network in 2015. Points represent binned data. Degree is $n + 1$, for depiction on a log axis; therefore farms with no connections, i.e. zero degree, are depicted as degree $= 1$.

when compared to random (electronic supplementary material, table S2), meaning that farms with low degree were connected to those with high degree, and vice versa, indicating that some farms act as 'hubs' for movements. The reciprocity of edges was 3–4 orders of magnitude higher in the observed network than that in the random network (electronic supplementary material, table S2), suggesting trading partners tend to reciprocate buying and selling cattle with one another. Clustering coefficients were higher, and average shortest paths were shorter, than in the random networks. GSCCs were smaller than in random networks, with the observed GSCCs containing fewer than half the number of farms of the random networks (electronic supplementary material, table S2). Together, this suggests that the observed networks are modular, consisting of multiple smaller groups of well-connected farms, and therefore displaying small-world-type properties.

Farm measures of movements remained consistent when they were ranked by the number of farms with which they traded (degree), and the number of cattle moved (strength), between years (electronic supplementary material, figure S4). This applied to trade both in-to and out-from the farm. Ranks of more global measures of connectivity for each farm (contact chains and betweenness) were also consistent between years but showed more variation than local measures (degree and strength) (electronic supplementary material, figure S4). These results indicate that individual farm movements tended to stay consistent over time.

Observed degree was highly variable, compared to the random network (electronic supplementary material, table S2). In- and out-degree of individual farms were positively skewed (skewness $= 17.5$ and 3.07, respectively; figure 3). Degree had power law exponents suggesting that the network might be characterized as scale-free, with many farms trading cattle with only a few direct partners and a small number of farms trading with many direct partners (electronic supplementary material, table S4). The number of premises from which individual farms buy-in cattle had much greater range (in-degree range $= 0–4346$) than the number of farms to which individual farms sell cattle (out-degree range $= 0–305$) (figure 4). The number of animals moved in-to and out-from individual farms also showed a positively skewed distribution and range that was more marked for the number of animals bought in (in-strength skewness $= 16.3$, range $= 0–15\,359$) than the number sold (out-strength skewness $= 8.67$, range $= 0–6,472$) (figure 4). These patterns for degree and strength were consistent across all years. Larger farms tended to trade with more other farms consistently among years, demonstrated by a positive relationship between herd size and degree (mean $r_s = 0.619$, 95% confidence interval 0.603–0.634). Larger farms also traded more animals (mean $r_s = 0.679$, 95% CI $= 0.662–0.696$). Herd size and out-degree and out-strength (mean out-degree $r_s = 0.531$, 95% CI $= 0.501–0.562$, s.e. $= 0.014$; mean out-strength $r_s = 0.587$, 95% CI $= 0.550–0.624$) were more strongly correlated than herd size and in-degree and in-strength (mean in-degree $r_s = 0.290$, 95% CI $= 0.280–0.300$; mean in-strength $r_s = 0.283$, 95% CI $= 0.274–0.292$). Differences between some herd types in the number of farms that were traded with, and number of animals traded, were clear. Median in-degree and in-strength were higher for fattening farms, and median out-degree and out-strength were higher for dairy farms (figure 4).

Betweenness values were positively skewed (skewness $= 29.2$), with a large group of farms showing low betweenness scores and a smaller proportion of farms with very high betweenness scores, but no

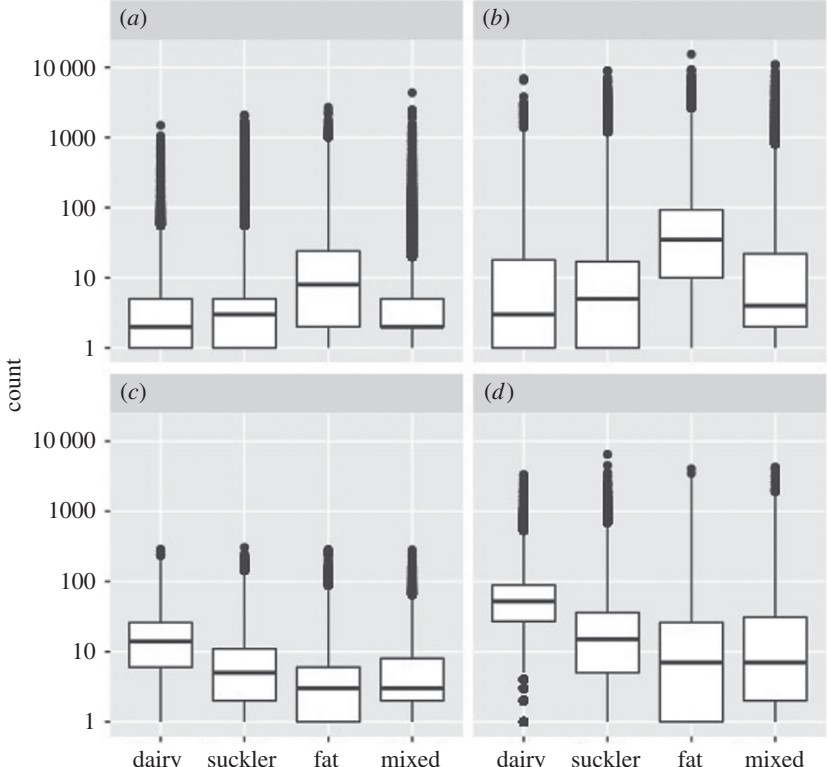

**Figure 4.** The number of farms directly traded with (degree) and the number of animals traded (strength) in the annual network split by herd type for: (*a*) in-degree (number of farms cattle purchased from), (*b*) in-strength (number of animals purchased), (*c*) out-degree (number of farms cattle sold to), (*d*) and out-strength (number of animals sold). Combined data from network analyses from 2001 to 2015 are shown. Data are $n + 1$, for depiction on a log axis, and include farms for which chain lengths were zero. Box plots show medians and interquartile ranges and the whiskers indicate the smallest or largest values no further than 1.5 times the interquartile ranges. Points outside this are outliers.

distinct differences between herd type (electronic supplementary material, figure S3). Mean betweenness was much larger in the observed network than in the randomized networks in all years but one, and median betweenness was much lower. This provides further evidence that some farms, not linked to a particular herd type, act as hubs within the network.

## 3.3. Contact chains

Most farms had fewer than 100 farms in their ICC; however, up to 40% of holdings had large or very large ICCs over a 1-year period, creating a strongly bimodal distribution (figure 5). The bi-modal distribution was also present for OCCs. The sizes of ICCs and OCCs for individual farms remained reasonably stable between all study years, though 2001 and 2002 have different characteristics (figure 6*c*,*d*). ICCs and OCCs were both positively skewed (skewness ingoing = 1.85, outgoing = 0.91). The maximum observed ICC of a single farm encompassed 86% of all British cattle holdings active in 2001 ($n = 73\,465$ farms) and the maximum OCC occurred in 2004, encompassing 43% ($n = 34\,460$ farms) of holdings. Across all years studied, approximately 50% of farms had very small ICCs, while 35–40% had large or very large ICCs (figure 5). More holdings in 2001 and 2002 have very small or small ICCs, than those from 2003 onwards (figure 6*c*). OCCs showed a different distribution; in most years, 50% of farms had very small to medium OCCs, and 50% had large or very large OCCs (figure 6*d*). More farms in 2001 and 2007 had large to very large OCCs, and in 2002, fewer farms had small or very small OCCs compared to other years (figure 6*d*). Clear differences were evident between herd types (figure 6*a*,*b*), with over 50% of fattening units but only 25% of breeding and mixed herds having large to very large ICCs (figure 6*a*). Over 80% of dairy farms, 55% of suckler farms and fewer than 40% of fattening farms had large to very large OCCs (figure 6*b*).

There was a weak correlation between ICC and OCC ($r_s = 0.181$, 95% CI = 0.174–0.187, $p < 0.001$, $n = 76\,031$) for the mean values calculated from 24 monthly spaced contact chains. This relationship

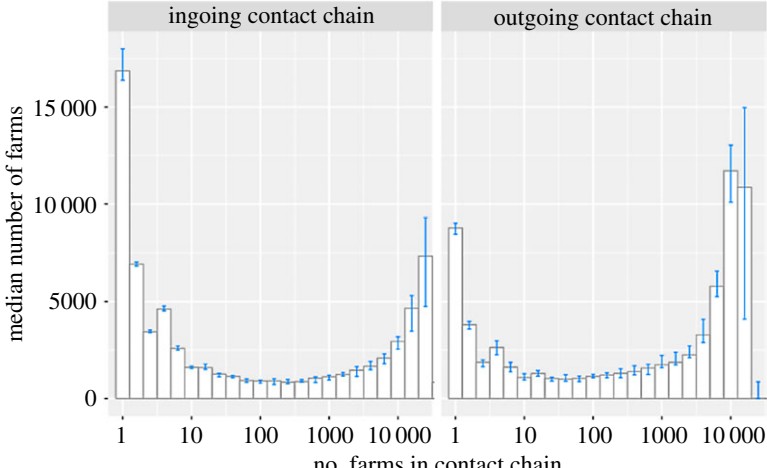

**Figure 5.** Distribution of ingoing and outgoing contact chains with the number of farms in the chain for: ingoing contact chains (ICCs) and outgoing contact chains (OCCs), showing bimodal distribution for both types of contact chain. ICCs have more farms with fewer farms in their chains, whereas despite a lower mean and maximum number of farms, OCCs have more farms with very large chains. Data are $n + 1$, for depiction on a log axis. Data include all active farms in Britain from 2001 to 2015. Error bars indicate the interquartile ranges.

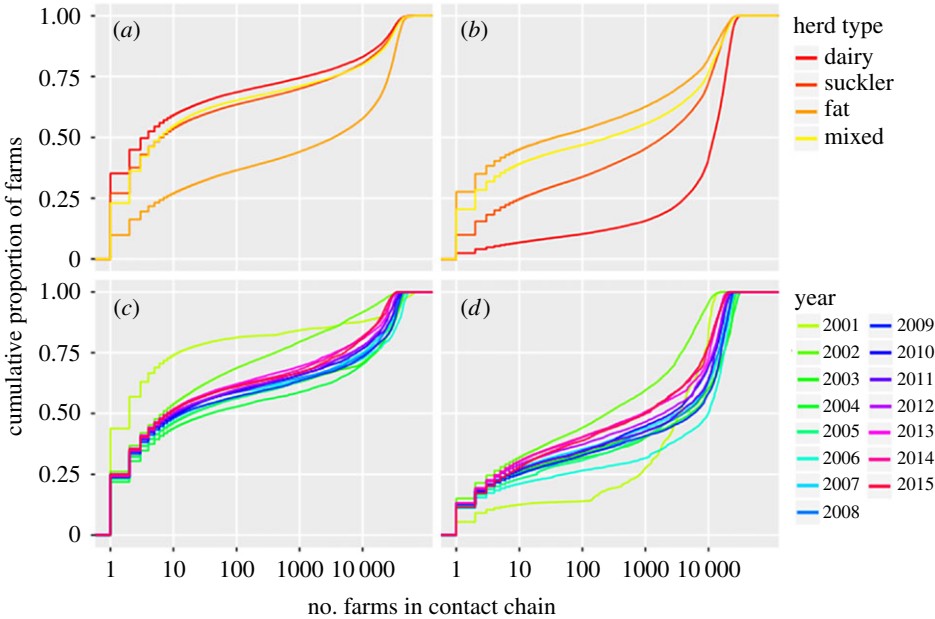

**Figure 6.** Annual cumulative proportions of farms in Britain by the number of farms in their contact chains 2001−2015 and by herd type. Contact chains from 2001 to 2015 are included; therefore, some farms are represented multiple times for herd types. All years show a similar distribution shape; however, there appears to be some drift over time and marked deviations for 2001 and 2002, probably due to FMDV movement restrictions, culling and restocking of herds in those years. Herd-type distributions have some similarities; however, fattening herds and dairy farms differ from other herd types for ICC and OCC, respectively. In (a,b), the colour of the line represents the herd type characterized by breed and sex CTS data, in (c,d), the colour of the line represents the year from 2001 to 2015. ICCs are shown in (a,c), OCCs are shown in (b,d). Data are $n + 1$, for depiction on a log axis; therefore farms with chain length of zero, are depicted as chain length = 1.

was weaker when correlations of ICC and OCC were compared within all study years (mean $r_s = 0.0398$, 95% CI = 0.020−0.594, $p < 0.001$ in all years, $n = 15$). Farms tended to cluster at small ICC and OCC, large ICC and OCC, and small ICC but large OCC (figure 7a). Regardless of ICC magnitude, dairy farms tended to have many farms in their OCC (figure 7b). Suckler herds tended to have more chains with large ICC and large OCC (figure 7c). Fattening herds had generally high ICCs with clustering at the low and high end of OCCs (figure 7d) and mixed herds tended to cluster more with low ICC and

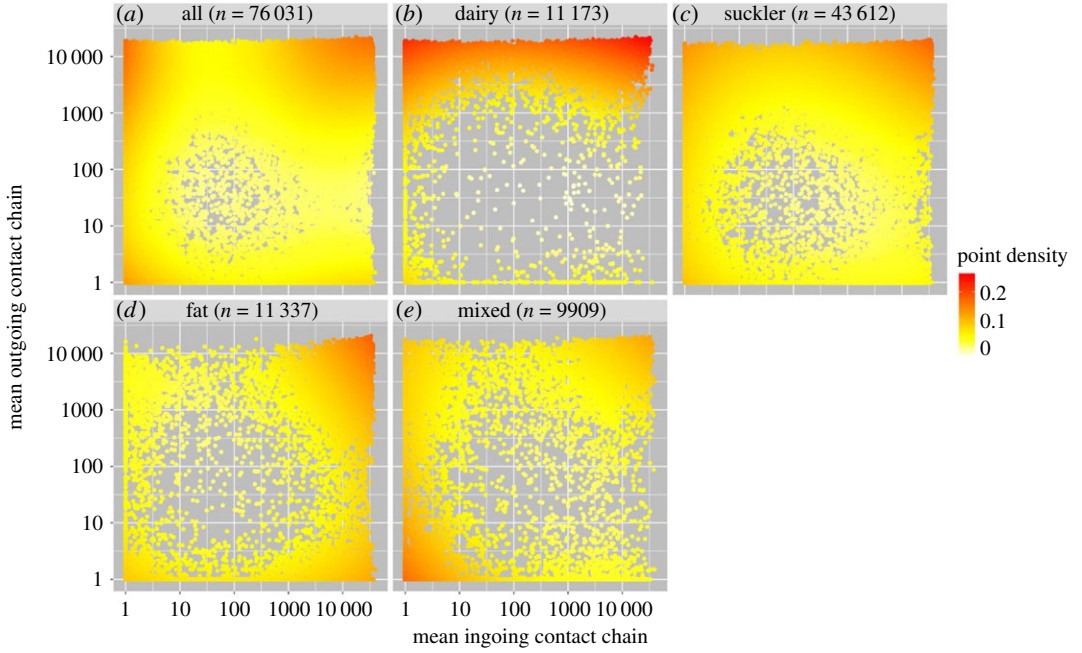

**Figure 7.** Point density scatterplots to show relationship between mean ICC and mean OCC, where many farms lie in the top right of the plot. Data are split by herd type (*a*) all, (*b*) dairy, (*c*) suckler, (*d*) fat and (*e*) mixed farms with mean values of contact chains using combined data from 24 sequential chains 2012 – 2014, showing that the distribution of point-density changes with herd type. Point density is shown on a colour scale with lines smoothed by local polynomial regression fitting with a span of 0.6. Data are $n + 1$, for depiction on log – log axes; therefore farms with chain length of zero are depicted as chain length = 1.

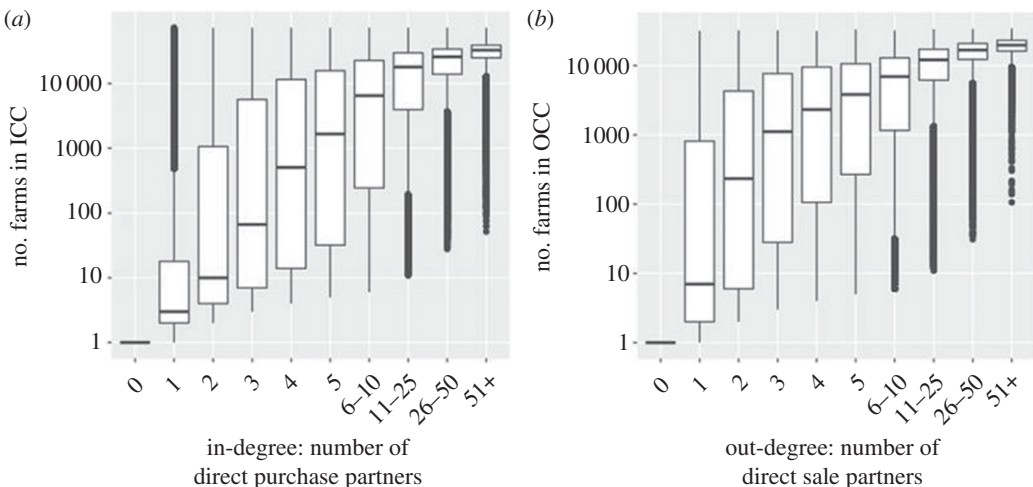

**Figure 8.** The relationship between degree and contact chains showing the increasing number of direct trading partners and increase in number of farms in contact chain. (*a*) In-degree and ingoing contact chains and (*b*) out-degree and outgoing contact chains. Chain lengths are from 2001 – 2015 and are $n + 1$, for depiction on a log axis. Box plots show medians and interquartile ranges and the whiskers indicate the smallest or largest values no further than 1.5 times the interquartile ranges. Points outside this are outliers.

OCC (figure 7*e*). There was a strong positive and consistent correlation between in-degree and annual ICC over all years from 2001 to 2015 (mean $r_s = 0.869$, 95% CI 0,858 – 0.879, $p < 0.001$ in all years). This was true to a lesser extent between OCC and out-degree (mean $r_s = 0.768$, 95% CI = 0.754 − 0.782, $p < 0.001$ in all years). Over three-quarters of farms that sold animals to between 6 and 10 farms in the study period had an OCC linking over 1000 farms, and farms that purchased animals from the same number of farms had a median ICC with 6486 farms. Some farms, despite low in- or out-degree, nevertheless had many farms in their contact chains (figure 8).

**Table 2.** Odds ratios and 95% confidence intervals for three logistic regression analyses to identify characteristics of farms with very large contact chains. The response variable is whether or not farms have (a) ICCs, (b) OCCs, and (c) ICCs *and* OCCs containing over 10 000 farms (values from combined 24 monthly-spaced chains). Herd size has been mean centred and is displayed for increments of 10 cattle (see electronic supplementary material, figure S7 for further herd size analysis).

| response variable | explanatory variable | levels | odds ratio | 2.50% | 97.50% |
|---|---|---|---|---|---|
| (a) ingoing contact chain greater than 10 000 | | (intercept) | 0.151 | 0.131 | 0.172 |
| | | herd size (per 10 cattle) | 1.004 | 1.002 | 1.006 |
| | herd type | dairy | Ref | | |
| | | fat | 9.509 | 8.659 | 10.454 |
| | | mixed | 1.280 | 1.162 | 1.411 |
| | | suckler | 1.281 | 1.189 | 1.381 |
| | region | east of England | Ref | | |
| | | east Midlands | 1.187 | 1.036 | 1.362 |
| | | northeast | 1.517 | 1.302 | 1.769 |
| | | northwest | 1.463 | 1.288 | 1.665 |
| | | Scotland | 0.767 | 0.677 | 0.872 |
| | | southeast | 0.297 | 0.251 | 0.350 |
| | | southwest | 0.577 | 0.509 | 0.656 |
| | | Wales | 0.484 | 0.425 | 0.553 |
| | | West Midlands | 1.073 | 0.942 | 1.225 |
| | | Yorkshire | 1.532 | 1.344 | 1.748 |
| | herd type: fat | herd size (per 10 cattle) | 1.186 | 1.176 | 1.196 |
| | herd type: mixed | herd size (per 10 cattle) | 1.052 | 1.047 | 1.057 |
| | herd type: suckler | herd size (per 10 cattle) | 1.046 | 1.043 | 1.049 |
| (b) outgoing contact chain greater than 10 000 | | (Intercept) | 0.211 | 0.161 | 0.272 |
| | | herd size (per 10 cattle) | 1.010 | 1.008 | 1.012 |
| | herd type | dairy | Ref | | |
| | | suckler | 0.093 | 0.087 | 0.100 |
| | | mixed | 0.112 | 0.101 | 0.123 |
| | | fat | 0.051 | 0.046 | 0.057 |
| | region | east of England | Ref | | |
| | | east Midlands | 1.526 | 1.153 | 2.047 |
| | | northeast | 11.584 | 8.817 | 15.442 |
| | | northwest | 18.075 | 14.009 | 23.716 |
| | | Scotland | 15.212 | 11.815 | 19.923 |
| | | southeast | 0.961 | 0.713 | 1.308 |
| | | southwest | 2.534 | 1.963 | 3.326 |
| | | Wales | 2.305 | 1.780 | 3.034 |
| | | West Midlands | 1.878 | 1.439 | 2.489 |
| | | Yorkshire | 3.646 | 2.796 | 4.828 |
| | herd type: fat | herd size (per 10 cattle) | 1.011 | 1.006 | 1.016 |
| | herd type: mixed | herd size (per 10 cattle) | 1.040 | 1.035 | 1.045 |
| | herd type: suckler | herd size (per 10 cattle) | 1.041 | 1.038 | 1.044 |

(*Continued.*)

| response variable | explanatory variable | levels | odds ratio | 2.50% | 97.50% |
|---|---|---|---|---|---|
| (*c*) ingoing and outgoing contact chains greater than 10 000 | | (Intercept) | 0.025 | 0.016 | 0.036 |
| | | herd size (per 10 cattle) | 1.003 | 1.001 | 1.006 |
| | herd type | dairy | *Ref* | | |
| | | fat | 0.493 | 0.433 | 0.561 |
| | | mixed | 0.496 | 0.428 | 0.574 |
| | | suckler | 0.372 | 0.335 | 0.412 |
| | region | east of England | *Ref* | | |
| | | east Midlands | 1.153 | 0.739 | 1.853 |
| | | northeast | 8.152 | 5.463 | 12.674 |
| | | northwest | 10.429 | 7.163 | 15.909 |
| | | Scotland | 6.360 | 4.371 | 9.696 |
| | | southeast | 0.383 | 0.215 | 0.678 |
| | | southwest | 2.263 | 1.545 | 3.468 |
| | | Wales | 1.197 | 0.801 | 1.863 |
| | | West Midlands | 2.156 | 1.447 | 3.347 |
| | | Yorkshire | 3.358 | 2.265 | 5.195 |
| | herd type: fat | herd size (per 10 cattle) | 1.020 | 1.014 | 1.025 |
| | herd type: mixed | herd size (per 10 cattle) | 1.031 | 1.025 | 1.036 |
| | herd type: suckler | herd size (per 10 cattle) | 1.034 | 1.030 | 1.037 |

Logistic regression indicated that fattening farms were overall 9.5 (95% CI 8.7–10.5) times more likely to have very large ICCs (greater than 10 000 farms in their ICC) than dairy farms (table 2). There were also regional differences, where farms in the north of England were more likely to have very large ICCs (table 2). Overall, dairy herds were more likely than any other herd type to have very large OCCs. However, with herd sizes above 500, mixed and suckler herds were more likely to have very large OCCs (electronic supplementary material, figure S7). Herds in the north of England and Scotland were much more likely to have very large OCCs than herds in the east of England (table 2). Herds with over 500 cattle had an increased likelihood of very large ICCs *and* OCCs, especially in non-dairy holdings (electronic supplementary material, figure S7). Again, herds in the north of England and Scotland were at higher risk of having very large ICCs *and* OCCs (table 2). ROC curve analysis showed all models had acceptable to excellent goodness of fit (electronic supplementary material, figure S8).

# 4. Discussion

Epidemics can be difficult to control if underlying transmission dynamics are not fully understood, especially in large networks where potential transmission pathways can be extensive and convoluted. The 9.5 million edges and approximately 70 000 nodes included in this study show that the British cattle network is complex, and potential transmission pathways can be extensive. Quantifying the extent of these chains is an important step in trying to understand the potential transmission routes for infections.

In respect of their contact chain distributions, British cattle holdings form two groups; those with very few (fewer than 10) contacts in their chains, and those with very many (more than 1000). Variation in this measure within a relatively short period could reflect important differences in a farm's risk of acquiring and spreading infection, and key opportunities for action at 'critical control points' in the network. Previous studies have reported similarly skewed data in annual cattle farm contact chains in Sweden and Switzerland [7,32], and ICCs of pig farms in Germany [53]. In our study, contact chains of individual farms were stable over time; however, larger chains showed some variation.

Large herd sizes are commonly associated with increased risk of disease [54] and this is often attributed to large numbers of animals being purchased [42], we found that although there is an overall positive relationship between herd size and the number of linked farms, numbers of animals traded and chain magnitude, these relationships varied among herd types. This suggests that there are other mechanisms, beyond more animals entering the herd, that contribute to the apparent increased risk of disease in large herds [55]. Dairy, suckler and fattening herds have distinct patterns of degree and contact chains, indicating that they play different roles in the network. Through the purchase of animals from many other farms, fattening herds may be more susceptible to acquiring infection, while being less likely to pass on infection via movements as, clearly, many fattening cattle move straight to slaughter. Through selling animals to many different farms, dairy farms that become infected may be disproportionately influential for disease spread in the cattle network, offering a potential target for control measures [53]. In addition to the established role of markets as a 'mixing pot' for highly transmissible diseases while animals are on site, here we emphasize their part in facilitating the dispersal of animals to many premises from one source farm, thereby potentially amplifying the spread of fast and slow-spreading diseases alike. In characterizing farms into only four groups, we inevitably simplify the diversity of cattle farming operations in the British cattle industry. Fattening farms represent animals most probably intended for beef production, but we do not distinguish between premises rearing animals from calves to slaughter weight and cattle 'dealers' purchasing store cattle to sell on to other dealers, markets or farms. These differing businesses may have varying impacts on disease dynamics, where dealers may exhibit properties similar to markets, in acting to disperse animals to many farms. Farms in the southeast and east of England, where cattle densities are lower [56], may be less well connected due to fewer chances to trade. Pre-movement bTB tests are not required for movement within the north of England and Scotland [57] and this may be responsible for the higher connectivity of their farms.

The cattle movement network in Britain displays scale-free properties, typical of those seen in movement networks in other countries. The number of cattle farms in Britain has decreased, driven largely by a reduction in numbers of smaller dairy farms and the formation of larger dairy herds (figure 1; electronic supplementary material, figure S2). However, similar numbers of cattle were traded between fewer farms over the study period, resulting in a substantial increase in network density. Movement restrictions for nine months of 2001, due to the FMDV outbreak [58], account for low numbers of separate movements (edges), smaller GSCC and low edge density that year (electronic supplementary material, table S2). However, the restocking of farms from larger batches of animals after resumption of movements gave rise to the larger number of cattle moved that year [37]. Although it is well documented that the cessation of bTB testing during 2001 [59] contributed to the spread of bTB [3,60], the increased volume of animals moved could have made a significant contribution to the subsequent increase and spread of bTB infection. The number of animals in a batch is likely to affect the risk of farms acquiring or transmitting infection [12], especially for pathogens with relatively low transmissibility such as bTB. Some studies have combined the number of animals traded with contact chains [21,32], and this could be incorporated in further analyses.

The density and clustering of the network are mid-range compared with other livestock movement networks [7,10,12,61], apart from an Italian network which had much lower clustering [9], suggesting British farms trade in small communities, and exhibit small-world properties. In the British network after 2004, clustering coefficients stayed stable and, after 2005, reciprocity decreased, suggesting that although the network becomes denser, it also becomes more dispersed, perhaps due to an increased propensity to travel further to trade cattle. The pattern of GSCC size in our results reflects that seen in previous studies [58,62], and the reduction in size seen from 2004 to 2009 is continued in our analysis, which extends to 2015. Reduction of the size of such key components has previously been associated with reduced risk of epidemics [15].

A small number of farms in our network act as hubs (nodes with many more direct trading partners than the majority). Previously, this network role was considered to be fulfilled predominantly by markets [62], hub farms provide similar linkages in the network that might facilitate epidemic spread by creating potential transmission 'shortcuts' through the network [63]. Negative degree assortativity, similar to Scandinavian networks, where direct sales between farms, rather than via markets, are the norm [7,61], combined with highly skewed degree distribution makes this type of network highly receptive to control measures targeted at hubs, rather than random selection [53]. It may therefore be beneficial to apply control measures, which have previously been aimed solely at markets, to hub farms as well. Risk-based trading measures might employ a proxy value for 'superspreader potential' that used network measures, similar to an 'infection potential' value [64], a probability of disease ratio [21], or

selecting a threshold for the most highly connected farms. This value would provide additional information on which farmers might base their buying decisions. However, it may be commercially harsh to rate farms in this way, as they have little control over this value beyond their direct purchases, and some farms even with very few direct trading partners are connected to very large contact chains (figure 8).

We used contact chains as temporally relevant network characteristics by which to assess the potential for acquiring and/or transmitting a slow-spreading infection arising from a farm's trading network. There are numerous methods by which to achieve similar proxy measures from analysis of movement networks, such as those used by Frössling et al. [21] and Büttner et al. [65]. The algorithms used by Rossi et al. [64] and Konschake et al. [66], allow a very fine scale of analysis, such that each node has its own infectious period and time of infection. We adopted a broader measure of contact chains, applied over an extended period of 15 years, to detect how variable farms may be in their network positions over time and in the context of infections with long incubation and infectious periods.

Although moving an infected animal into the herd presents the clearest risk of disease transmission, farms have other 'connections', including neighbouring farms and contacts via fomites, service providers and wildlife, any of which might be important for transmission of infections [67–69]. These connections often occur at a local scale and have varying importance depending on the pathogen of interest. Animal movements, however, can be implicated at all spatial scales, from 43% of movements which occur within 20 km of the source farm to the substantial number of long-range movements documented in Britain [70]. Animal movements between neighbouring farms are also likely to be underestimated due to local practices [71]. Here, we have focused on this movement of animals as a potential pathway for the transmission of chronic infections and suggest that due to the long timescale, infections transmitted via movements may be more extensive through the network than we expect and that investigations into direct contacts may not be sufficient to trace the source and reach of some infections. Thus more extensive contact chains may better guide us to some of the varied sources and transmission pathways of cattle infections.

By examining networks of cattle movements, we observed two distinct patterns of interaction; many farms quickly became connected to a large proportion of the national network, yet some remained relatively isolated. We have shown marked variation between farms, not only in degree and betweenness but also in the more complex contact chains among British cattle farms, sustained over a period of 15 years, that has been characterized by change in the industry and recovery from catastrophic disease outbreaks. Farms that exhibit extremes in ICCs and OCCs fit with expectations of 'superspreaders' of infection [72]. The British networks' scale-free type properties suggest that the industry may benefit from targeted control of these influential nodes [53].

Risks associated with direct trading partners are relatively easy for farmers to consider. However, the chains to which they can become connected remain hidden, along with the potential risks of exposure to infection they bring. By increasing the number of their direct trading partners, farmers are likely to see large, and sometimes very large, increases in the number of farms in their contact chain. Chains can also quickly become large, even with very few direct contacts, perhaps leaving farmers who believe they trade 'carefully' with a false sense of security. Knowledge of contact chains, and the trading patterns and history of the farms from which they are buying, might better equip farmers to judge the exposure associated with their animal trading behaviour. Contact chains allow us to assess a farm's role within the network and further investigation should explore their application to target certain farms, herd types or practices for improved control of diseases.

Data accessibility. Further information and figures supporting this article have been uploaded as part of the electronic supplementary material. Underlying data consist of every movement of cattle between all farms in Great Britain. Aside from the size of the dataset, there are substantial issues of confidentiality (locations, trading practices) and commercial sensitivity in these data. They are collated and managed by Defra, via the Animal and Plant Health Agency, who grant access to the data with specific permissions for specific studies. In practice, this means that the data can be used for the stated purpose only, and making the data publicly accessible would not conform to the licence the authors have been granted to these data. With the agreement of the journal's Editorial Office, the authors will not be able to make the dataset available on this occasion, but encourage readers, referees and editors to contact the Animal and Plant Health Agency data manager for data access requests. At the time of submission, the data manager is Andy Mitchell (andrew.mitchell@apha.gsi.gov.uk).

Authors' contributions. All authors conceived the study, H.R.F. carried out data analysis, T.J.M. and M.J.S. assisted with data manipulation, H.R.F. and R.A.M. drafted the manuscript. All authors reviewed the manuscript and gave final approval for publication.

Competing interests. The authors have no competing interests.

**Funding.** H.R.F. was funded by an Industrial CASE studentship from the BBSRC grant no. BB/M015874/1, in partnership with the APHA.

**Acknowledgements.** The authors gratefully acknowledge Stefan Widgren for computational assistance, Sara Downs and reviewers for their comments on earlier drafts, and the APHA data and epidemiology teams for providing data and assisting with data interpretation.

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
