## [Reviewer comments · Royal Society Open Science]

Review History

RSOS-180719.R0 (Original submission)

Review form: Reviewer 1 (Susanna Sternberg Lewerin)

Is the manuscript scientifically sound in its present form?

Yes

Are the interpretations and conclusions justified by the results?

Yes

Is the language acceptable?

Yes

Is it clear how to access all supporting data?

Yes

Do you have any ethical concerns with this paper?

No

Have you any concerns about statistical analyses in this paper?

No

Recommendation?

Accept with minor revision (please list in comments)

Comments to the Author(s)

The manuscript deals with an important aspect of animal disease spread by a relevant approach. However, as the movement of animals via markets is a phenomenon that plays a much more important role in the British isles than elsewhere in Europe, it would be useful for the reader to include some additional text about this. For some diseases, animals meeting at markets/shows (despite biosecurity measures in place) constitutes a major risk of dissemination and it would therefore be of great interest to see all the analyses performed on the dataset with markets and shows included as well as excluded. As this is relevant for 35% of the movements (p5, 146) it is not unimportant. At least, Figure S1 in the supplementary material could be placed next to Figure 2 to illustrate this, and some text added in relevant sections (Methods, Results, Discussion). Moreover, as bovine TB is referred to (and rightly so), it would be of interest to relate the movement patterns (e.g. the pattern of the GSCC) to the different geographical risk areas for TB or, at least, add some comments about this in the Discussion.

I have some minor comments, listed below:

P7, L8: directed or direct?

P12, L 36-41. The comparison with Sweden needs an added statement that the Swedish cattle trading system doesn't include livestock markets, the similarity is a large number of animals into fattening farms, not a proportion of these via livestock markets. In addition, Swedish fattening herds only sell directly to slaughter.

P13, L14-15: "these" farms, not this farms

P14, L20: rather "than" random selection

P14, L27-28: perhaps point out that moving infected animals remains the highest risk (other contacts <100%)

P15 L 9: it appears that you are hinting that contacts and ICC information should be added to farms' disease status information, this might be pointed out more clearly in the discussion, with pro's and con's of such a system.

In addition, the following points would be useful to add to the Discussion:

"Fattening herds" in most countries (except for UK and Ireland) sell only animals directly to slaughter, in this study it is demonstrated that some have large ICCs AND large OCCs, is it possible to characterise these more (where are they, are they really "fattening" herds or something else)?

As TB is mentioned as a disease for which the study is relevant, the finding that animal movements increased while TB testing almost stopped entirely is worth commenting on. In many papers the stopped testing has been put forward as the main reason for subsequent increase in TB prevalence and animal movements have not been highlighted. This study demonstrates the importance of movements and it would be a shame not to emphasise it. The many animal movements still resulted in a lower number of edges in 2001, this should also be commented on, the particular aspects of restocking depopulated herds could be elaborated.

Some herds with low in-degree and out-degree still had large ICCs and OCCs, this is also worth commenting on (false sense of security?)

High ICC's in the North of England is also worth mentioning in connection with TB, as a lot of the low risk area for TB is in the North.

Review form: Reviewer 2

Is the manuscript scientifically sound in its present form?

No

Are the interpretations and conclusions justified by the results?

No

Is the language acceptable?

Yes

Is it clear how to access all supporting data?

No

Do you have any ethical concerns with this paper?

No

Have you any concerns about statistical analyses in this paper?

No

Recommendation?

Major revision is needed (please make suggestions in comments)

Comments to the Author(s)

The paper titled "Contact chains of cattle farms in Great Britain" describes the cattle movement network in the study country during a time span of 15 years (2001-2015). The authors applied several network analysis metric in order to characterize the 15 yearly networks. Particular attention was given to a metric called "contact chain" (in the literature known also as "infection chain"), and to the merit of the authors this metrics was rarely used on such extensive datasets, in particular in GB (to my knowledge).

The paper has the clear merit to conduct a complete network analysis, including many metrics that can describe the between-farms contact patterns due to animal movements, and to conduct this research through a long time-series. However, novelty elements hardly emerge from both the methods and the discussion/conclusion, since they applied metrics developed by others and already used in the context of livestock networks, and the conclusions are rather well-known in the literature. In fact, the major concern regards the structure and the lack of a sound scientific question to drive the paper.

Major comments

Lack of scientific question drive

Overall, the present study lack of a clear question-answer drive and, in the current form, looks more like an application of a network analysis measures. This lack is reflected in the structure of the Introduction and Discussion sections. While the former is confusing and does not follow a clear logical flow, the latter seems to partly repeat the results section (except for few paragraphs, such as page 12 the 1st and 2nd, and page 13, the 3rd), and to partly to state already known. I think that a descriptive analysis might be still valuable, but only if better justified and well described.

Contact chains calculation

The second major comment regard the calculation of contact chains. The paragraph reported from Page 7 line 28 to Page 8 line 5 is confusing and it took me a lot to understand how authors calculated the contact chains.

In my understanding, they calculate the ICC (and OCC) starting each year from January for 365

days. My concern is that by calculating the contact chains in this way, they overestimate the chains value. For example, if a given farm (#2) receive an animal in February from farm #1, then is inactive for a prolonged time period, and in November sends an animal to farm #3, are farm 1-2-3 in the same chain? Kanschake et al (2014, PlosOne) introduced an infection period for the calculation of contacts chains to overcome this issue. In this formulation two contacts needs to happen within the infectious period to be included in the same chain. The authors formulation could work in the context of slow-spreading diseases (i.e. bTB) in the absence of control measures, such as movement restrictions, but not for fast-spreading diseases (i.e. FMD). Another issue with that is the seasonality of movements and “border effect”. The authors showed that by starting the calculation in different months (over a range of 3 years) results did not changed much. However, this sounds obvious, because the periods with more movements will be caught anyway by the chain, as cattle movement are known to have a marked yearly seasonal behaviour. So, I do not think this issue is addressed correctly. On the other hand, a “border effect” could prevent a correct calculation of contact chains. As an example, if I start to calculate the ICC on Jan 1st of a given year and ignore what happened in the last period of the previous year. How do the authors addressed this issue? Finally, in order to combine the information of the ICC and OCC the authors could use the method showed by Rossi et al (2017, Plos Computational Biology), or at least discuss their results in the light of this. In general, I would refresh the literature on contact chains.

Minor comments

General

- The citations numbering is not correct (example, page 4 and 5, no 36 and 38 between 35 and 39, 37 is the same as 33). Also, sometimes citations are redundant (see page 3, line 14).
- Networks measures are calculated for each year (so, 15 observations per measure, in my understanding): most of the time the authors reports mean and SD. I think it would be better to report the median/mean and 95th percentile or median/mean and range.

Introduction

Page 3

Lines 6-8: add citation;

Line 9: “animal movements between farms form a directional link from the source to the destination farm”, it is how we consider movements in a network framework, more than the movement forming an edge itself;

Lines 25-27: add citation;

Lines 27-30: support this claim;

Lines 49-50: “cows”? Or “farms” maybe;

Methods

Page 5

There is no brief description of the different types of farms. A reader unfamiliar with the GB cattle industry might not be aware of the exact difference between the types of farm. Also, is “suckler” what elsewhere is called “cow-calf”?

Lines 48-50: I agree with this choice, but it needs to be properly justified and/or discussed;

Pag 6

Lines 15-20: the choice of weight the links based on the number of animals and not on the number of batches is good to me, but it should be supported or discussed (see Volkova et al., 2010, already cited);

Line 30: update igragh citation;

Lines 15-56 (and Page 7 lines 3-24): I would suggest the authors to report all these network metrics, with oportune citations, and brief descriptions in a table, since none of these represent a novelty in the field. Maybe this could make the section easier to read;

Page 7

Lines 5-10: the correct citation here is Kao et al., 2006, Interface.

Page 8

Lines 7-19: how much the results of this analysis can be affected by a different threshold choice? I would suggest the authors to repeat the analysis with at least 2 alternative thresholds.

Alternatively, they can use a GLM with multinomial response variable. In any case, this choice needs more explanation and discussion;

Line 12: I would suggest to show a map with the England's regions in the supplementary material, since they are using this information;

Results

Page 8

Could be interested to report the number of isolated farms every year, if they change or not.

Line 55: what does "batched movements" mean?

Page 9

Line 33/Figure S5: should the bars be vertical on this figure? Otherwise they necessitate a better explanation;

Page 10

Lines 4-20: this paragraph is very confusing. Please rephrase;

Line 36: quantify "few";

Page 11

Lines 48-50: I could not find the ROC curve;

Discussion

Page 12

Lines 15-17: quantify "few" and "very many";

Lines 31-41: as described, these results seem trivial;

Page 13

Lines 12-21: this paragraph looks disconnected, please rephrase;

Lines 23-25: add citation, and maybe fully test the power-law distribution on the degree centrality (see Clauset, Shalizi, Newman, 2009, SIAM Review paper);

Page 14

Lines 27-33: instead of "widespread" it could be more precise to talk about the different spatial scales at which other types of contacts might be important, and for which kind of infections;

Conclusion

Page 14

Lines 39-56: this paragraph should be included in the Introduction.

Decision letter (RSOS-180719.R0)

17-Sep-2018

Dear Dr McDonald,

The editors assigned to your paper ("Contact chains of cattle farms in Great Britain") have now received comments from reviewers. We would like you to revise your paper in accordance with the referee and Associate Editor suggestions which can be found below (not including confidential reports to the Editor). Please note this decision does not guarantee eventual acceptance.

Please submit a copy of your revised paper before 10-Oct-2018. Please note that the revision deadline will expire at 00.00am on this date. If we do not hear from you within this time then it will be assumed that the paper has been withdrawn. In exceptional circumstances, extensions may be possible if agreed with the Editorial Office in advance. We do not allow multiple rounds of revision so we urge you to make every effort to fully address all of the comments at this stage. If deemed necessary by the Editors, your manuscript will be sent back to one or more of the original reviewers for assessment. If the original reviewers are not available, we may invite new reviewers.

- Data accessibility

<http://datadryad.org/submit?journalID=RSOS&manu=RSOS-180719>

- Competing interests

- Authors' contributions

All submissions, other than those with a single author, must include an Authors' Contributions section which individually lists the specific contribution of each author. The list of Authors

should meet all of the following criteria; 1) substantial contributions to conception and design, or acquisition of data, or analysis and interpretation of data; 2) drafting the article or revising it critically for important intellectual content; and 3) final approval of the version to be published.

- Acknowledgements

- Funding statement

Please note that Royal Society Open Science charge article processing charges for all new submissions that are accepted for publication. Charges will also apply to papers transferred to Royal Society Open Science from other Royal Society Publishing journals, as well as papers submitted as part of our collaboration with the Royal Society of Chemistry (<http://rsos.royalsocietypublishing.org/chemistry>). If your manuscript is newly submitted and subsequently accepted for publication, you will be asked to pay the article processing charge, unless you request a waiver and this is approved by Royal Society Publishing. You can find out more about the charges at <http://rsos.royalsocietypublishing.org/page/charges>. Should you have any queries, please contact openscience@royalsociety.org.

Kind regards,

Royal Society Open Science Editorial Office
Royal Society Open Science
openscience@royalsociety.org

on behalf of Dr Dirk Drasdo (Associate Editor) and Prof. Kevin Padian (Subject Editor)
openscience@royalsociety.org

Comments to Author:

Reviewers' Comments to Author:

Reviewer: 1

Comments to the Author(s)

The manuscript deals with an important aspect of animal disease spread by a relevant approach. However, as the movement of animals via markets is a phenomenon that plays a much more important role in the British isles than elsewhere in Europe, it would be useful for the reader to

include some additional text about this. For some diseases, animals meeting at markets/shows (despite biosecurity measures in place) constitutes a major risk of dissemination and it would therefore be of great interest to see all the analyses performed on the dataset with markets and shows included as well as excluded. As this is relevant for 35% of the movements (p5, 146) it is not unimportant. At least, Figure S1 in the supplementary material could be placed next to Figure 2 to illustrate this, and some text added in relevant sections (Methods, Results, Discussion). Moreover, as bovine TB is referred to (and rightly so), it would be of interest to relate the movement patterns (e.g. the pattern of the GSCC) to the different geographical risk areas for TB or, at least, add some comments about this in the Discussion.

I have some minor comments, listed below:

P7, L8: directed or direct?

P12, L 36-41. The comparison with Sweden needs an added statement that the Swedish cattle trading system doesn't include livestock markets, the similarity is a large number of animals into fattening farms, not a proportion of these via livestock markets. In addition, Swedish fattening herds only sell directly to slaughter.

P13, L14-15: "these" farms, not this farms

P14, L20: rather "than" random selection

P14, L27-28: perhaps point out that moving infected animals remains the highest risk (other contacts <100%)

P15 L 9: it appears that you are hinting that contacts and ICC information should be added to farms' disease status information, this might be pointed out more clearly in the discussion, with pro's and con's of such a system.

In addition, the following points would be useful to add to the Discussion:

"Fattening herds" in most countries (except for UK and Ireland) sell only animals directly to slaughter, in this study it is demonstrated that some have large ICCs AND large OCCs, is it possible to characterise these more (where are they, are they really "fattening" herds or something else?)

As TB is mentioned as a disease for which the study is relevant, the finding that animal movements increased while TB testing almost stopped entirely is worth commenting on. In many papers the stopped testing has been put forward as the main reason for subsequent increase in TB prevalence and animal movements have not been highlighted. This study demonstrates the importance of movements and it would be a shame not to emphasise it. The many animal movements still resulted in a lower number of edges in 2001, this should also be commented on, the particular aspects of restocking depopulated herds could be elaborated.

Some herds with low in-degree and out-degree still had large ICCs and OCCs, this is also worth commenting on (false sense of security?)

High ICC's in the North of England is also worth mentioning in connection with TB, as a lot of the low risk area for TB is in the North.

Reviewer: 2

Comments to the Author(s)

The paper titled "Contact chains of cattle farms in Great Britain" describes the cattle movement network in the study country during a time span of 15 years (2001-2015). The authors applied several network analysis metric in order to characterize the 15 yearly networks. Particular attention was given to a metric called "contact chain" (in the literature known also as "infection chain"), and to the merit of the authors this metrics was rarely used on such extensive datasets, in particular in GB (to my knowledge).

The paper has the clear merit to conduct a complete network analysis, including many metrics that can describe the between-farms contact patterns due to animal movements, and to conduct this research through a long time-series. However, novelty elements hardly emerge from both the

methods and the discussion/conclusion, since they applied metrics developed by others and already used in the context of livestock networks, and the conclusions are rather well-known in the literature. In fact, the major concern regards the structure and the lack of a sound scientific question to drive the paper.

Major comments

Lack of scientific question drive

Overall, the present study lack of a clear question-answer drive and, in the current form, looks more like an application of a network analysis measures. This lack is reflected in the structure of the Introduction and Discussion sections. While the former is confusing and does not follow a clear logical flow, the latter seems to partly repeat the results section (except for few paragraphs, such as page 12 the 1st and 2nd, and page 13, the 3rd), and to partly to state already known. I think that a descriptive analysis might be still valuable, but only if better justified and well described.

Contact chains calculation

The second major comment regard the calculation of contact chains. The paragraph reported from Page 7 line 28 to Page 8 line 5 is confusing and it took me a lot to understand how authors calculated the contact chains.

In my understanding, they calculate the ICC (and OCC) starting each year from January for 365 days. My concern is that by calculating the contact chains in this way, they overestimate the chains value. For example, if a given farm (#2) receive an animal in February from farm #1, then is inactive for a prolonged time period, and in November sends an animal to farm #3, are farm 1-2-3 in the same chain? Korschake et al (2014, PlosOne) introduced an infection period for the calculation of contacts chains to overcome this issue. In this formulation two contacts needs to happen within the infectious period to be included in the same chain. The authors formulation could work in the context of slow-spreading diseases (i.e. bTB) in the absence of control measures, such as movement restrictions, but not for fast-spreading diseases (i.e. FMD).

Another issue with that is the seasonality of movements and "border effect". The authors showed that by starting the calculation in different months (over a range of 3 years) results did not changed much. However, this sounds obvious, because the periods with more movements will be caught anyway by the chain, as cattle movement are known to have a marked yearly seasonal behaviour. So, I do not think this issue is addressed correctly. On the other hand, a "border effect" could prevent a correct calculation of contact chains. As an example, if I start to calculate the ICC on Jan 1st of a given year and ignore what happened in the last period of the previous year. How do the authors addressed this issue?

Finally, in order to combine the information of the ICC and OCC the authors could use the method showed by Rossi et al (2017, Plos Computational Biology), or at least discuss their results in the light of this. In general, I would refresh the literature on contact chains.

Minor comments

General

- The citations numbering is not correct (example, page 4 and 5, no 36 and 38 between 35 and 39, 37 is the same as 33). Also, sometimes citations are redundant (see page 3, line 14).
- Networks measures are calculated for each year (so, 15 observations per measure, in my understanding): most of the time the authors reports mean and SD. I think it would be better to report the median/mean and 95th percentile or median/mean and range.

Introduction

Page 3

Lines 6-8: add citation;

Line 9: “animal movements between farms form a directional link from the source to the destination farm”, it is how we consider movements in a network framework, more than the movement forming an edge itself;

Lines 25-27: add citation;

Lines 27-30: support this claim;

Lines 49-50: “cows”? Or “farms” maybe;

Methods

Page 5

There is no brief description of the different types of farms. A reader unfamiliar with the GB cattle industry might not be aware of the exact difference between the types of farm. Also, is “suckler” what elsewhere is called “cow-calf”?

Lines 48-50: I agree with this choice, but it needs to be properly justified and/or discussed;

Page 6

Lines 15-20: the choice of weight the links based on the number of animals and not on the number of batches is good to me, but it should be supported or discussed (see Volkova et al., 2010, already cited);

Line 30: update igraph citation;

Lines 15-56 (and Page 7 lines 3-24): I would suggest the authors to report all these network metrics, with opportune citations, and brief descriptions in a table, since none of these represent a novelty in the field. Maybe this could make the section easier to read;

Page 7

Lines 5-10: the correct citation here is Kao et al., 2006, Interface.

Page 8

Lines 7-19: how much the results of this analysis can be affected by a different threshold choice? I would suggest the authors to repeat the analysis with at least 2 alternative thresholds.

Alternatively, they can use a GLM with multinomial response variable. In any case, this choice needs more explanation and discussion;

Line 12: I would suggest to show a map with the England’s regions in the supplementary material, since they are using this information;

Results

Page 8

Could be interested to report the number of isolated farms every year, if they change or not.

Line 55: what does “batched movements” mean?

Page 9

Line 33/Figure S5: should the bars be vertical on this figure? Otherwise they necessitate a better explanation;

Page 10

Lines 4-20: this paragraph is very confusing. Please rephrase;

Line 36: quantify “few”;

Page 11

Lines 48-50: I could not find the ROC curve;

Discussion

Page 12

Lines 15-17: quantify “few” and “very many”;

Lines 31-41: as described, these results seem trivial;

Page 13

Lines 12-21: this paragraph looks disconnected, please rephrase;

Lines 23-25: add citation, and maybe fully test the power-law distribution on the degree centrality (see Clauset, Shalizi, Newman, 2009, SIAM Review paper);

Page 14

Lines 27-33: instead of “widespread” it could be more precise to talk about the different spatial scales at which other types of contacts might be important, and for which kind of infections;

Conclusion

Page 14

Lines 39-56: this paragraph should be included in the Introduction.

Author's Response to Decision Letter for (RSOS-180719.R0)

See Appendix A.

RSOS-180719.R1 (Revision)

Review form: Reviewer 1 (Susanna Sternberg Lewerin)

Is the manuscript scientifically sound in its present form?

Yes

Are the interpretations and conclusions justified by the results?

Yes

Is the language acceptable?

Yes

Is it clear how to access all supporting data?

Yes

Do you have any ethical concerns with this paper?

No

Have you any concerns about statistical analyses in this paper?

No

Recommendation?

Accept as is

Comments to the Author(s)

I have only one remaining question, that does not really qualify as a revision.

I would suggest to double-check the statement on line 216-217 that 5.4% of fattening herds were closed for the five-year period (2011-2015). Is it reasonable that about 450 herds (if I understand the figures correctly) that are classified as fattening herds would not introduce a single animal for 5 years but still qualify as active herds (so there must be outward movements as births would not be expected to occur at any significant rate in fattening herds). A brief explanatory comment could be added to this sentence, as it does sound extraordinary to have any number of "closed" fattening herds for a longer period.

Review form: Reviewer 2

Is the manuscript scientifically sound in its present form?

Yes

Are the interpretations and conclusions justified by the results?

Yes

Is the language acceptable?

Yes

Is it clear how to access all supporting data?

Not Applicable

Do you have any ethical concerns with this paper?

No

Have you any concerns about statistical analyses in this paper?

No

Recommendation?

Accept with minor revision (please list in comments)

Comments to the Author(s)

The quality how this manuscript has improved and the authors were able to effectively clarify the purpose of it. These efforts made the manuscript publishable.

However, I would suggest a couple of things. First, to make more clear (in the abstract, in particular) that OCC and ICCs calculations are suited to bovine TB. In general, the abstract needs a bit more explanation on the background and the significance of the paper (as an example, one important result to me is that even after excluding markets, some farms act as "hubs" anyway). Second, I would also suggest to fix the syntax of some sentences, to improve the readability and the flow of the manuscript (example: lines 19-20 or 84-85, but throughout the text). Other minor comments below.

Line 36: "pathogens"?

Line 42: influence on what?

Lines 98-107: this paragraph is already explaining some methods, maybe it needs a better collocation

Lines 160-170: I might have intuitively understood what you did exactly to address the "border effect", but this part is not clear. Please, rephrase.

Line 182: "and" is bold

Line 386: in my understanding the ones cited here are not "transmission models" but rather algorithms to calculate the contact chains

Line 419: the method is at least 10 years old (first Dube' paper was published in 2008)

Decision letter (RSOS-180719.R1)

04-Jan-2019

Dear Dr McDonald:

On behalf of the Editors, I am pleased to inform you that your Manuscript RSOS-180719.R1 entitled "Contact chains of cattle farms in Great Britain" has been accepted for publication in Royal Society Open Science subject to minor revision in accordance with the referee suggestions. Please find the referees' comments at the end of this email.

The reviewers and Subject Editor have recommended publication, but also suggest some minor revisions to your manuscript. Therefore, I invite you to respond to the comments and revise your manuscript.

- Ethics statement

- Data accessibility

<http://datadryad.org/submit?journalID=RSOS&manu=RSOS-180719.R1>

- Competing interests

- Authors' contributions

AB carried out the molecular lab work, participated in data analysis, carried out sequence alignments, participated in the design of the study and drafted the manuscript; CD carried out the statistical analyses; EF collected field data; GH conceived of the study, designed the study,

coordinated the study and helped draft the manuscript. All authors gave final approval for publication.

- Acknowledgements

- Funding statement

Because the schedule for publication is very tight, it is a condition of publication that you submit the revised version of your manuscript before 13-Jan-2019. Please note that the revision deadline will expire at 00.00am on this date. If you do not think you will be able to meet this date please let me know immediately.

Supplementary files will be published alongside the paper on the journal website and posted on the online figshare repository (<https://figshare.com>). The heading and legend provided for each supplementary file during the submission process will be used to create the figshare page, so please ensure these are accurate and informative so that your files can be found in searches. Files

on figshare will be made available approximately one week before the accompanying article so that the supplementary material can be attributed a unique DOI.

on behalf of Dr Dirk Drasdo (Associate Editor) and Kevin Padian (Subject Editor)
openscience@royalsociety.org

Associate Editor Comments to Author (Dr Dirk Drasdo):

Associate Editor: 1

Comments to the Author:

Dear Authors,

I thank you for your constructive responses on the suggestions of the reviewers. I think this outcome was an example for an excellent interplay between authors and reviewers resulting in a considerable improvement of the paper quality.

I finally like you to consider the minor modifications suggested by one of the reviewers, which look to me aiming at even further improving the manuscript and its perception. I leave it to you in how far you like to modify the text in response to the reviewers' suggestions.

With kind regards,
Dirk Drasdo

Reviewer comments to Author:

Reviewer: 1

Comments to the Author(s)

I have only one remaining question, that does not really qualify as a revision.

I would suggest to double-check the statement on line 216-217 that 5.4% of fattening herds were closed for the five-year period (2011-2015). Is it reasonable that about 450 herds (if I understand the figures correctly) that are classified as fattening herds would not introduce a single animal for 5 years but still qualify as active herds (so there must be outward movements as births would not be expected to occur at any significant rate in fattening herds). A brief explanatory comment could be added to this sentence, as it does sound extraordinary to have any number of "closed" fattening herds for a longer period.

Reviewer: 2

Comments to the Author(s)

The quality how this manuscript has improved and the authors were able to effectively clarify the purpose of it. These efforts made the manuscript publishable.

However, I would suggest a couple of things. First, to make more clear (in the abstract, in particular) that OCC and ICCs calculations are suited to bovine TB. In general, the abstract needs a bit more explanation on the background and the significance of the paper (as an example, one important result to me is that even after excluding markets, some farms act as “hubs” anyway). Second, I would also suggest to fix the syntax of some sentences, to improve the readability and the flow of the manuscript (example: lines 19-20 or 84-85, but throughout the text). Other minor comments below.

Line 36: “pathogens”?

Line 42: influence on what?

Lines 98-107: this paragraph is already explaining some methods, maybe it needs a better collocation

Lines 160-170: I might have intuitively understood what you did exactly to address the “border effect”, but this part is not clear. Please, rephrase.

Line 182: “and” is bold

Line 386: in my understanding the ones cited here are not “transmission models” but rather algorithms to calculate the contact chains

Line 419: the method is at least 10 years old (first Dube’ paper was published in 2008)

Author's Response to Decision Letter for (RSOS-180719.R1)

See Appendix B.

Decision letter (RSOS-180719.R2)

23-Jan-2019

Dear Dr McDonald,

I am pleased to inform you that your manuscript entitled "Contact chains of cattle farms in Great Britain" is now accepted for publication in Royal Society Open Science.

Royal Society Open Science operates under a continuous publication model (<http://bit.ly/cpFAQ>). Your article will be published straight into the next open issue and this will be the final version of the paper. As such, it can be cited immediately by other researchers.

As the issue version of your paper will be the only version to be published I would advise you to check your proofs thoroughly as changes cannot be made once the paper is published.

on behalf of Dr Dirk Drasdo (Associate Editor) and Professor Kevin Padian (Subject Editor)
openscience@royalsociety.org

Appendix A

Contact chains of cattle farms in Great Britain

Helen R. Fielding, Trevelyan J. McKinley, Matthew J. Silk, Richard J. Delahay and
Robbie A. McDonald

Response to reviewers

Reviewer 1

1. The manuscript deals with an important aspect of animal disease spread by a relevant approach. However, as the movement of animals via markets is a phenomenon that plays a much more important role in the British isles than elsewhere in Europe, it would be useful for the reader to include some additional text about this. For some diseases, animals meeting at markets/shows (despite biosecurity measures in place) constitutes a major risk of dissemination and it would therefore be of great interest to see all the analyses performed on the dataset with markets and shows included as well as excluded. As this is relevant for 35% of the movements (p5, l46) it is not unimportant. At least, Figure S1 in the supplementary material could be placed next to Figure 2 to illustrate this, and some text added in relevant sections (Methods, Results, Discussion).
 - As requested, we have moved Figure S1 into the main document (now Figure 2b) and added clarification and explanation to the figure legend (Line 669), methods (Lines 121-129), results (207-209), and discussion (332-335, 339-341, and particularly 368-376). Our analysis reveals and emphasises the importance of some farms as hubs, in addition to the previously described role of markets. This effect would be less apparent in an analysis that included markets as nodes.
 - See also our response to Referee 2 (Point 27), who agrees with our approach of directly linking source and destination farms.
2. Moreover, as bovine TB is referred to (and rightly so), it would be of interest to relate the movement patterns (e.g. the pattern of the GSCC) to the different geographical risk areas for TB or, at least, add some comments about this in the Discussion.
 - We fully agree with this comment, however extending the analysis to a direct assessment of TB risk is a major addition that would extend the length and scope of this particular paper too far. We hope you will appreciate that we are therefore dealing with the issue of TB, and the risk areas, more extensively in a separate paper.

I have some minor comments, listed below:

3. P7, L8: directed or direct?
 - This text is now in supplementary Table S1, as per point 30 of Reviewer 2 (below), in the row about the Giant Strongly-Connected Component.
 - We use 'directed' to indicate a difference between the strong and weak components, whereas the weak component would include farms that are connected by an edge in either direction (direct), the strong component specifically uses only farms linked by edges in a determined direction (directed).

4. P12, L 36-41. The comparison with Sweden needs an added statement that the Swedish cattle trading system doesn't include livestock markets, the similarity is a large number of animals into fattening farms, not a proportion of these via livestock markets. In addition, Swedish fattening herds only sell directly to slaughter.
 - Done as requested (Line 372).

5. P13, L14-15: "these" farms, not this farms
 - Sentence no longer exists. These lines have been rephrased in response to Reviewer 2.

6. P14, L20: rather "than" random selection
 - Done as requested (Line 374).

7. P14, L27-28: perhaps point out that moving infected animals remains the highest risk (other contacts <100%)
 - Done as requested (Line 391)

8. P15 L 9: it appears that you are hinting that contacts and ICC information should be added to farms' disease status information, this might be pointed out more clearly in the discussion, with pro's and con's of such a system.
 - We now discuss this further at Lines 376-382 and again at Lines 412-421.

9. In addition, the following points would be useful to add to the Discussion: "Fattening herds" in most countries (except for UK and Ireland) sell only animals directly to slaughter, in this study it is demonstrated that some have large ICCs AND large OCCs, is it possible to characterise these more (where are they, are they really "fattening" herds or something else?)?
 - We have clarified the classification of herd types at Lines 134-139 and its limitations at lines 335-341.
 - We looked at location of different types and found that fattening premises were well-distributed across GB (see the map below), across all cattle dense areas, therefore did not consider the spatial distribution was worth reporting in its own right.

10. As TB is mentioned as a disease for which the study is relevant, the finding that animal movements increased while TB testing almost stopped entirely is worth commenting on. In many papers the stopped testing has been put forward as the main reason for subsequent increase in TB prevalence and animal movements have not been highlighted. This study demonstrates the importance of movements and it would be a shame not to emphasise it. The many animal movements still resulted in a lower number of edges in 2001, this should also be commented on, the particular aspects of restocking depopulated herds could be elaborated.

- We have added text at lines 349-355.

11. Some herds with low in-degree and out-degree still had large ICCs and OCCs, this is also worth commenting on (false sense of security?)

- We now address this at lines 298-299 in the results, and Figure 8, and have added text in the discussion at lines 379-382 and 412-419.

12. High ICC's in the North of England is also worth mentioning in connection with TB, as a lot of the low risk area for TB is in the North.

- We agree but refer you to our response to Point 2 above. This will be in a later paper.

Reviewer: 2

The paper titled "Contact chains of cattle farms in Great Britain" describes the cattle movement network in the study country during a time span of 15 years (2001-2015). The authors applied several network analysis metric in order to characterize

the 15 yearly networks. Particular attention was given to a metric called “contact chain” (in the literature known also as “infection chain”), and to the merit of the authors this metrics was rarely used on such extensive datasets, in particular in GB (to my knowledge).

The paper has the clear merit to conduct a complete network analysis, including many metrics that can describe the between-farms contact patterns due to animal movements, and to conduct this research through a long time-series. However, novelty elements hardly emerge from both the methods and the discussion/conclusion, since they applied metrics developed by others and already used in the context of livestock networks, and the conclusions are rather well-known in the literature. In fact, the major concern regards the structure and the lack of a sound scientific question to drive the paper.

Major comments

13. Lack of scientific question drive

Overall, the present study lack of a clear question-answer drive and, in the current form, looks more like an application of a network analysis measures. This lack is reflected in the structure of the Introduction and Discussion sections. While the former is confusing and does not follow a clear logical flow, the latter seems to partly repeat the results section (except for few paragraphs, such as page 12 the 1st and 2nd, and page 13, the 3rd), and to partly to state already known. I think that a descriptive analysis might be still valuable, but only if better justified and well described.

- We have revisited parts of the introduction in order to improve logical flow, which we now consider to be clear, i.e. introducing the importance of network analysis in general and in terms of control measures > variation arising from pathogen characteristics and the time-scale of infectiousness > time-ordered networks > some detail of such network analyses as they pertain to cattle > a discussion of the situation in Britain > the specific context, highlighting the focus on chronic, slow-moving or latent infections and bTB > predictions arising.
- We have ironed out repetition, changed perspectives and consolidated extensive sections of the Discussion.
- We agree that the paper is an application of network measures to a complete network of particular interest and over an extended period. Patterns of variation and change in this network potentially have great epidemiological and economic importance. We feel this study makes a worthwhile contribution to understanding disease transmission in a sector where cattle movements have become so numerous and far-reaching.

14. Contact chains calculation

The second major comment regard the calculation of contact chains. The paragraph reported from Page 7 line 28 to Page 8 line 5 is confusing and it took me a lot to understand how authors calculated the contact chains.

- We have rewritten and clarified the contact chains methodology (lines 156-179). A schematic and further detail are provided in the Supplementary Materials.
15. In my understanding, they calculate the ICC (and OCC) starting each year from January for 365 days. My concern is that by calculating the contact chains in this way, they overestimate the chains value. For example, if a given farm (#2) receive an animal in February from farm #1, then is inactive for a prolonged time period, and in November sends an animal to farm #3, are farm 1-2-3 in the same chain? Kunschake et al (2014, PlosOne) introduced an infection period for the calculation of contacts chains to overcome this issue. In this formulation two contacts needs to happen within the infectious period to be included in the same chain. The authors formulation could work in the context of slow-spreading diseases (i.e. bTB) in the absence of control measures, such as movement restrictions, but not for fast-spreading diseases (i.e. FMD).
- We have clarified our approach in the Introduction at Lines 52-59, 71-83 and Discussion 383-390, and to emphasise that, in line with the reviewer, we used a one year period to represent the length of time a slow-spreading infection, such as bTB, may have come onto a farm before being identified and control measures being implemented.
16. Another issue with that is the seasonality of movements and “border effect”. The authors showed that by starting the calculation in different months (over a range of 3 years) results did not changed much. However, this sounds obvious, because the periods with more movements will be caught anyway by the chain, as cattle movement are known to have a marked yearly seasonal behaviour. So, I do not think this issue is addressed correctly. On the other hand, a “border effect” could prevent a correct calculation of contact chains. As an example, if I start to calculate the ICC on Jan 1st of a given year and ignore what happened in the last period of the previous year. How do the authors addressed this issue?
- We have clarified our methods at lines 160-170 to highlight our evaluation of a 36 month period and how it resolves problems associated with a border effect.
17. Finally, in order to combine the information of the ICC and OCC the authors could use the method showed by Rossi et al (2017, Plos Computational Biology), or at least discuss their results in the light of this.
- We have clarified in the methods at lines 180-191. We have included a discussion on our techniques in light of the Rossi paper at lines 377, 386-390. Our classification of those farms with over 10,000 farms in both their ingoing and outgoing contact chain was our proxy for a ‘superspreader’ farm.
18. In general, I would refresh the literature on contact chains.

- We have added relevant citations from recent literature throughout the introduction and discussion.

Minor comments

General

19. The citations numbering is not correct (example, page 4 and 5, no 36 and 38 between 35 and 39, 37 is the same as 33). Also, sometimes citations are redundant (see page 3, line 14).
 - We have corrected these errors.
20. Networks measures are calculated for each year (so, 15 observations per measure, in my understanding): most of the time the authors reports mean and SD. I think it would be better to report the median/mean and 95th percentile or median/mean and range.
 - We have changed standard deviation to 95% confidence intervals when reporting these network measures throughout the results.

Introduction

Page 3

21. Lines 6-8: add citation;
 - Citation added at Line 37.
22. Line 9: “animal movements between farms form a directional link from the source to the destination farm”, it is how we consider movements in a network framework, more than the movement forming an edge itself;
 - Updated at Lines 37-40.
23. Lines 25-27: add citation;
 - This line has now been removed in response to previous comments regarding the introduction.
24. Lines 27-30: support this claim;
 - Added citation at line 52.
25. Lines 49-50: “cows”? Or “farms” maybe;
 - Text added at lines 61-65 to clarify that the networks were specifically constructed from individual adult dairy cow movements between farms.

Methods

Page 5

26. There is no brief description of the different types of farms. A reader unfamiliar with the GB cattle industry might not be aware of the exact difference between the types of farm. Also, is “suckler” what elsewhere is called “cow-calf”?
 - We have added in descriptions of the different herds at lines 134-139.

27. Lines 48-50: I agree with this choice, but it needs to be properly justified and/or discussed;
- Added text at lines 121-129 for further discussion and justification. See also our response to Referee 1, point 1, above.

Page 6

28. Lines 15-20: the choice of weight the links based on the number of animals and not on the number of batches is good to me, but it should be supported or discussed (see Volkova et al., 2010, already cited);
- Text added at lines 143-145 and Volkova et al. (2010) are cited as [15].
29. Line 30: update igraph citation;
- We have updated the version number and used the citation provided in the R documentation.
30. Lines 15-56 (and Page 7 lines 3-24): I would suggest the authors to report all these network metrics, with opportune citations, and brief descriptions in a table, since none of these represent a novelty in the field. Maybe this could make the section easier to read;
- We have created a new Table S1 providing these definitions.

Page 7

31. Lines 5-10: the correct citation here is Kao et al., 2006, Interface.
- This text is now included in Table S1. The best paper is 'Kiss IZ, Green DM, Kao RR. 2006 The network of sheep movements within Great Britain: Network properties and their implications for infectious disease spread. *J. R. Soc. Interface* **3**, 669–677. (doi:10.1098/rsif.2006.0129)' and has been added to the references in the Supplementary Materials.

Page 8

32. Lines 7-19: how much the results of this analysis can be affected by a different threshold choice? I would suggest the authors to repeat the analysis with at least 2 alternative thresholds. Alternatively, they can use a GLM with multinomial response variable. In any case, this choice needs more explanation and discussion;
- We have repeated the analysis as requested with three thresholds; 100, 1000, and 10,000 and included the ROC values (Table S2) and predicted probabilities for herd size and farm type in the supplementary material (Figures S5-7). We have added text explaining this in the methods at lines 183-185.
 - The threshold of 10,000 was used for the main body of the paper as it had the highest ROC values for ICCs, OCCs and both contact chains. We defined by number of farms rather than a percentile as this encompassed very different numbers of farms for ICCs and OCCs, due to the differences in

distribution between ICCs and OCCs. On balance, we felt it was easier to interpret with clear farm numbers than proportions.

33. Line 12: I would suggest to show a map with the England's regions in the supplementary material, since they are using this information;
- As suggested, we have added a map into the supplementary materials as Figure S10 and referred to at line 188.

Results

Page 8

34. Could be interested to report the number of isolated farms every year, if they change or not.
- We have added this at Lines 210-217.
35. Line 55: what does "batched movements" mean?
- We have removed this term here but clarified at lines 118 and again at 219-221.

Page 9

36. Line 33/Figure S5: should the bars be vertical on this figure? Otherwise the necessitate a better explanation;
- We have completely reformatted the figure (now Figure S4).

Page 10

37. Lines 4-20: this paragraph is very confusing. Please rephrase;
- Text rephrased as advised.
38. Line 36: quantify "few";
- Clarified at line 270.

Page 11

39. Lines 48-50: I could not find the ROC curve;
- Added in as Figure S7.

Discussion

Page 12

40. Lines 15-17: quantify "few" and "very many";
- Clarified at line 318.
41. Lines 31-41: as described, these result seem trivial;
- We have reduced this to a brief mention at Line 332.

Page 13

42. Lines 12-21: this paragraph looks disconnected, please rephrase;
- Rephrased and relocated to line 346.

43. Lines 23-25: add citation, and maybe fully test the power-law distribution on the degree centrality (see Clauset, Shalizi, Newman, 2009, SIAM Review paper);
- We have added this citation and referred to the definition in the new Table S1. We calculated the power law for overall degree centrality (Table S4). We compared a power-law model for the distribution to a log-normal model, though there is no evidence that either fitted better (Table S4 legend).

Page 14

44. Lines 27-33: instead of “widespread” it could be more precise to talk about the different spatial scales at which other types of contacts might be important, and for which kind of infections;
- We have added text and clarified at lines 393-400.

Conclusion

Page 14

45. Lines 39-56: this paragraph should be included in the Introduction.
- We have moved this to the Introduction.

Appendix B

Associate Editor Comments to Author (Dr Dirk Drasdo):

Associate Editor: 1

Comments to the Author:

Dear Authors,

I thank you for your constructive responses on the suggestions of the reviewers. I think this outcome was an example for an excellent interplay between authors and reviewers resulting in a considerable improvement of the paper quality.

I finally like you to consider the minor modifications suggested by one of the reviewers, which look to me aiming at even further improving the manuscript and its perception. I leave it to you in how far you like to modify the text in response to the reviewers' suggestions.

With kind regards,

Dirk Drasdo

Thank you. Here are our responses.

Reviewer comments to Author:

Reviewer: 1

Comments to the Author(s)

I have only one remaining question, that does not really qualify as a revision.

I would suggest to double-check the statement on line 216-217 that 5.4% of fattening herds were closed for the five-year period (2011-2015). Is it reasonable that about 450 herds (if I understand the figures correctly) that are classified as fattening herds would not introduce a single animal for 5 years but still qualify as active herds (so there must be outward movements as births would not be expected to occur at any significant rate in fattening herds). A brief explanatory comment could be added to this sentence, as it does sound extraordinary to have any number of "closed" fattening herds for a longer period.

Text clarified at Lines 238-240)

Reviewer: 2

Comments to the Author(s)

The quality how this manuscript has improved and the authors were able to effectively clarify the purpose of it. These efforts made the manuscript publishable.

However, I would suggest a couple of things. First, to make more clear (in the abstract, in particular) that OCC and ICCs calculations are suited to bovine TB.

In general, the abstract needs a bit more explanation on the background and the significance of the paper (as an example, one important result to me is that even after excluding markets, some farms act as "hubs" anyway).

Second, I would also suggest to fix the syntax of some sentences, to improve the readability and the flow of the manuscript (example: lines 19-20 or 84-85, but throughout the text).

We have rephrased several sentences in the abstract.

Changed lines 84-85 (now at lines 89-90).

Minor changes to improve readability throughout.

Other minor comments below.

Line 36: “pathogens”?

Changed ‘infections’ to ‘pathogens’ as requested (line 42).

Line 42: influence on what?

Streamlined text (line 44).

Lines 98-107: this paragraph is already explaining some methods, maybe it needs a better collocation

We have looked at this (lines 102-109) and consider that the text is appropriate to an Introduction and stops well short of an account of the Methods.

Lines 160-170: I might have intuitively understood what you did exactly to address the “border effect”, but this part is not clear. Please, rephrase.

Added text for clarification (line 170-175).

Line 182: “and” is bold

Corrected (line 180)

Line 386: in my understanding the ones cited here are not “transmission models” but rather algorithms to calculate the contact chains

Corrected (line 390)

Line 419: the method is at least 10 years old (first Dube’ paper was published in 2008)

Corrected (line 423)